# ENHANCE WORD REPRESENTATION FOR OUT-OF-VOCABULARY ON UBUNTU DIALOGUE CORPUS

## ABSTRACT

Ubuntu dialogue corpus is the largest public available dialogue corpus to make it feasible to build end-to-end deep neural network models directly from the conversation data. One challenge of Ubuntu dialogue corpus is the large number of out-of-vocabulary words. In this paper we proposed a method which combines the general pre-trained word embedding vectors with those generated on the task-specific training set to address this issue. We integrated character embedding into Chen et al's Enhanced LSTM method (ESIM) and used it to evaluate the effectiveness of our proposed method. For the task of next utterance selection, the proposed method has demonstrated a significant performance improvement against original ESIM and the new model has achieved state-of-the-art results on both Ubuntu dialogue corpus and Douban conversation corpus. In addition, we investigated the performance impact of end-of-utterance and end-of-turn token tags.

## 1 INTRODUCTION

The ability for a machine to converse with human in a natural and coherent manner is one of challenging goals in AI and natural language understanding. One problem in chat-oriented human-machine dialog system is to reply a message within conversation contexts. Existing methods can be divided into two categories: retrieval-based methods (Wang et al., 2013; Ji et al., 2014; Yan et al., 2016b) and generation based methods (Vinyals & Le, 2015). The former is to rank a list of candidates and select a good response. For the latter, encoder-decoder framework (Vinyals & Le, 2015) or statistical translation method (Ritter et al., 2011) are usually used to generate a response. It is not easy to main the fluency of the generated texts.

Ubuntu dialogue corpus (Lowe et al., 2015) is the public largest unstructured multi-turns dialogue corpus which consists of about one-million two-person conversations. The size of the corpus makes it attractive for the exploration of deep neural network modeling in the context of dialogue systems. Most deep neural networks use word embedding as the first layer. They either use fixed pre-trained word embedding vectors generated on a large text corpus or learn word embedding for the specific task. The former is lack of flexibility of domain adaptation. The latter requires a very large training corpus and significantly increases model training time. Word out-of-vocabulary issue occurs for both cases. Ubuntu dialogue corpus also contains many technical words (e.g. "ctrl+alt+f1", "/dev/sdb1"). The ubuntu corpus (V2) contains 823057 unique tokens whereas only 22% tokens occur in the pre-built GloVe word vectors[1]. Although character-level representation which models sub-word morphologies can alleviate this problem to some extent (Huang et al., 2013; Bojanowski et al., 2016; Kim et al., 2016), character-level representation still have limitations: learn only morphological and orthographic similarity, other than semantic similarity (e.g. 'car' and 'bmw') and it cannot be applied to Asian languages (e.g. Chinese characters).

In this paper, we generate word embedding vectors on the training corpus based on word2vec (Mikolov et al., 2013). Then we propose an algorithm to combine the generated one with the pre-trained word embedding vectors on a large general text corpus based on vector concatenation. The new word representation maintains information learned from both general text corpus and task-domain. The nice property of the algorithm is simplicity and little extra computational cost will be added. It can address word out-of-vocabulary issue effectively. This method can be applied to most

---

[1] glove.42B.300d.zip in https://nlp.stanford.edu/projects/glove/

NLP deep neural network models and is language-independent. We integrated our methods with ESIM(baseline model) (Chen et al., 2017). The experimental results have shown that the proposed method has significantly improved the performance of original ESIM model and obtained state-of-the-art results on both Ubuntu Dialogue Corpus and Douban Conversation Corpus (Wu et al., 2017). On Ubuntu Dialogue Corpus (V2), the improvement to the previous best baseline model (single) on $R_{10}@1$ is 3.8% and our ensemble model on $R_{10}@1$ is 75.9%. On Douban Conversation Corpus, the improvement to the previous best model (single) on $P@1$ is 3.6%.

Our contributions in this paper are summarized below:

1. We propose an algorithm to combine pre-trained word embedding vectors with those generated on the training corpus to address out-of-vocabulary word issues and experimental results have shown that it is very effective.

2. ESIM with our method has achieved the state-of-the-art results on both Ubuntu Dialogue corpus and Douban conversation corpus.

3. We investigate performance impact of two special tags on Ubuntu Dialogue Corpus: end-of-utterance and end-of-turn.

The rest paper is organized as follows. In Section 2, we review the related work. In Section 3 we provide an overview of ESIM (baseline) model and describe our methods to address out-of-vocabulary issues. In Section 4, we conduct extensive experiments to show the effectiveness of the proposed method. Finally we conclude with remarks and summarize our findings and outline future research directions.

## 2 RELATED WORK

Character-level representation has been widely used in information retrieval, tagging, language modeling and question answering. Shen et al. (2014) represented a word based on character trigram in convolution neural network for web-search ranking. Bojanowski et al. (2016) represented a word by the sum of the vector representation of character n-gram. Santos et al (Santos & Zadrozny, 2014; Santos & Guimaraes, 2015) and Kim et al. (2016) used convolution neural network to generate character-level representation (embedding) of a word. The former combined both word-level and character-level representation for part-of-speech and name entity tagging tasks while the latter used only character-level representation for language modeling. Yang et al. (2016b) employed a deep bidirectional GRU network to learn character-level representation and then concatenated word-level and character-level representation vectors together. Yang et al. (2016a) used a fine-grained gating mechanism to combine the word-level and character-level representation for reading comprehension. Character-level representation can help address out-of-vocabulary issue to some extent for western languages, which is mainly used to capture character ngram similarity.

The other work related to enrich word representation is to combine the pre-built embedding produced by GloVe and word2vec with structured knowledge from semantic network ConceptNet (Speer & Havasi, 2012) and merge them into a common representation (Speer & Chin, 2016). The method obtained very good performance on word-similarity evaluations. But it is not very clear how useful the method is for other tasks such as question answering. Furthermore, this method does not directly address out-of-vocabulary issue.

Next utterance selection is related to response selection from a set of candidates. This task is similar to ranking in search, answer selection in question answering and classification in natural language inference. That is, given a context and response pair, assign a decision score (Baudiš et al., 2016). Ji et al. (2014) formalized short-text conversations as a search problem where rankSVM was used to select response. The model used the last utterance (a single-turn message) for response selection. On Ubuntu dialogue corpus, Lowe et al. (2015) proposed Long Short-Term Memory(LSTM) (Hochreiter & Schmidhuber, 1997) siamese-style neural architecture to embed both context and response into vectors and response were selected based on the similarity of embedded vectors. Kadlec et al. (2015) built an ensemble of convolution neural network (CNN) (Kim, 2014) and Bi-directional LSTM. Baudiš et al. (2016) employed a deep neural network structure (Tan et al., 2015) where CNN was applied to extract features after bi-directional LSTM layer. Zhou et al. (2016) treated each turn in multi-turn context as an unit and joined word sequence view and utterance sequence view together

by deep neural networks. Wu et al. (2017) explicitly used multi-turn structural info on Ubuntu dialogue corpus to propose a sequential matching method: match each utterance and response first on both word and sub-sequence levels and then aggregate the matching information by recurrent neural network.

The latest developments have shown that attention and matching aggregation are effective in NLP tasks such as question/answering and natural language inference. Seo et al. (2016) introduced context-to-query and query-to-context attentions mechanisms and employed bi-directional LSTM network to capture the interactions among the context words conditioned on the query. Parikh et al. (2016) compared a word in one sentence and the corresponding attended word in the other sentence and aggregated the comparison vectors by summation. Chen et al. (2017) enhanced local inference information by the vector difference and element-wise product between the word in premise an the attended word in hypothesis and aggregated local matching information by LSTM neural network and obtained the state-of-the-art results on the Stanford Natural Language Inference (SNLI) Corpus. Wang et al. (2017) introduced several local matching mechanisms before aggregation, other than only word-by-word matching.

## 3   OUR MODEL

In this section, we first review ESIM model (Chen et al., 2017) and introduce our modifications and extensions. Then we introduce a string matching algorithm for out-of-vocabulary words.

### 3.1   ESIM MODEL

In our notation, given a context with multi-turns $C = (c_1, c_2, \cdots, c_i, \cdots, c_m)$ with length $m$ and a response $R = (r_1, r_2, \cdots, r_j, \cdots, r_n)$ with length $n$ where $c_i$ and $r_j$ is the $i$th and $j$th word in context and response, respectively. For next utterance selection, the response is selected based on estimating a conditional probability $P(y = 1|C, R)$ which represents the confidence of selecting $R$ from the context $C$. Figure 1 shows high-level overview of our model and its details will be explained in the following sections.

**Word Representation Layer**. Each word in context and response is mapped into $d$-dimensional vector space. We construct this vector space with word-embedding and character-composed embedding. The character-composed embedding, which is newly introduced here and was not part of the original forumulation of ESIM, is generated by concatenating the final state vector of the forward and backward direction of bi-directional LSTM (BiLSTM). Finally, we concatenate word embedding and character-composed embedding as word representation.

**Context Representation Layer**. As in base model, context and response embedding vector sequences are fed into BiLSTM. Here BiLSTM learns to represent word and its local sequence context. We concatenate the hidden states at each time step for both directions as local context-aware new word representation, denoted by $\bar{a}$ and $\bar{b}$ for context and response, respectively.

$$\bar{a}_i = \text{BiLSTM}(\bar{a}_{i-1}, w_i),\ 1 \leq i \leq m, \tag{1}$$

$$\bar{b}_i = \text{BiLSTM}(\bar{b}_{j-1}, w_j),\ 1 \leq j \leq n, \tag{2}$$

where $w$ is word vector representation from the previous layer.

**Attention Matching Layer**. As in ESIM model, the co-attention matrix $E \in \mathbb{R}^{m \times n}$ where $E_{ij} = \bar{a}_i^T \bar{b}_j$. $E_{ij}$ computes the similarity of hidden states between context and response. For each word in context, we find the most relevant response word by computing the attended response vector in Equation 3. The similar operation is used to compute attended context vector in Equation 4.

$$\tilde{a}_i = \sum_{j=1}^{n} \frac{\exp(E_{ij})}{\sum_{k=1}^{n} \exp(E_{ik})} \bar{b}_j,\ 1 \leq i \leq m, \tag{3}$$

$$\tilde{b}_j = \sum_{i=1}^{m} \frac{\exp(E_{ij})}{\sum_{k=1}^{m} \exp(E_{kj})} \bar{a}_i,\ 1 \leq j \leq n. \tag{4}$$

After the above attended vectors are calculated, vector difference and element-wise product are used to enrich the interaction information further between context and response as shown in Equation 5

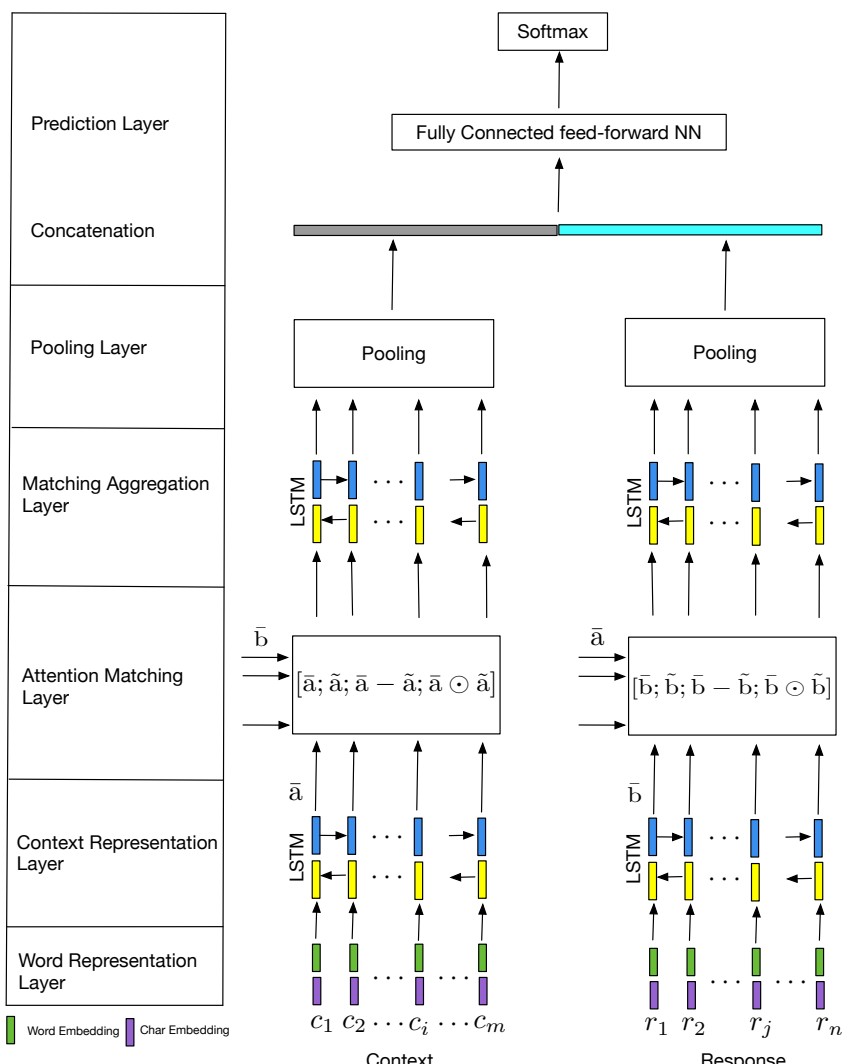

Figure 1: A high-level overview of ESIM layout. Compared with the original one in (Chen et al., 2017), the diagram addes character-level embedding and replaces average pooling by LSTM last state summary vector.

and 6.

$$\text{m}_i^a = [\bar{a}_i; \tilde{a}_i; \bar{a}_i - \tilde{a}_i; \bar{a}_i \odot \tilde{a}_i], \ 1 \le i \le m, \tag{5}$$

$$\text{m}_j^b = [\bar{b}_j; \tilde{b}_j; \bar{b}_j - \tilde{b}_j; \bar{b}_j \odot \tilde{b}_j], \ 1 \le j \le n, \tag{6}$$

where the difference and element-wise product are concatenated with the original vectors.

**Matching Aggregation Layer**. As in ESIM model, BiLSTM is used to aggregate response-aware context representation as well as context-aware response representation. The high-level formula is given by

$$\text{v}_i^a = \text{BiLSTM}(\text{v}_{i-1}^a, m_i^a), \ 1 \le i \le m, \tag{7}$$

$$\text{v}_j^b = \text{BiLSTM}(\text{v}_{j-1}^b, m_j^b), \ 1 \le j \le n. \tag{8}$$

**Pooling Layer**. As in ESIM model, we use max pooling. Instead of using average pooling in the original ESIM model, we combine max pooling and final state vectors (concatenation of both

forward and backward one) to form the final fixed vector, which is calculated as follows:

$$v_{max}^a = \max_{i=1}^{m} v_i^a, \tag{9}$$

$$v_{max}^b = \max_{j=1}^{n} v_j^b, \tag{10}$$

$$v = [v_{max}^a; v_{max}^b; v_m^a; v_n^b]. \tag{11}$$

**Prediction Layer**. We feed $v$ in Equation 11 into a 2-layer fully-connected feed-forward neural network with ReLu activation. In the last layer the sigmoid function is used. We minimize binary cross-entropy loss for training.

## 3.2 METHODS FOR OUT-OF-VOCABULARY

Many pre-trained word embedding vectors on general large text-corpus are available. For domain-specific tasks, out-of-vocabulary may become an issue. Here we propose algorithm 1 to combine pre-trained word vectors with those word2vec (Mikolov et al., 2013) generated on the training set. Here the pre-trainined word vectors can be from known methods such as GloVe (Pennington et al., 2014), word2vec (Mikolov et al., 2013) and FastText (Bojanowski et al., 2016).

---

**Algorithm 1:** Combine pre-trained word embedding with those generated on training set.

---

**Input** : Pre-trained word embedding set $\{U_w | w \in S\}$ where $U_w \in \mathbb{R}^{d_1}$ is embedding vector for word $w$. Word embedding $\{V_w | w \in T\}$ are generated on training set where $V_w \in \mathbb{R}^{d_2}$. P is a set of word vocabulary on the task dataset.

**Output:** A dictionary with word embedding vectors of dimension $d_1 + d_2$ for $(S \cap P) \cup T$.

res = dict()
 **for** $w \in (S \cap P) \cup T$ **do**
   **if** $w \in S \cap P$ and $w \in T$ **then** res[$w$] = $[U_w; V_w]$;
   **else if** $w \in S \cap P$ and $w \notin T$ **then** res[$w$] = $[U_w; \vec{0}]$;
   **else** res[$w$] = $[\vec{0}; V_w]$;
**end**
**Return** res

---

where $[;]$ is vector concatenation operator. The remaining words which are in $P$ and are not in the above output dictionary are initialized with zero vectors. The above algorithm not only alleviates out-of-vocabulary issue but also enriches word embedding representation.

## 4 EXPERIMENT

### 4.1 DATASET

We evaluate our model on the public Ubuntu Dialogue Corpus V2 [2] (Lowe et al., 2017) since this corpus is designed for response selection study of multi turns human-computer conversations. The corpus is constructed from Ubuntu IRC chat logs. The training set consists of 1 million $< context, response, label >$ triples where the original context and corresponding response are labeled as positive and negative response are selected randomly on the dataset. On both validation and test sets, each context contains one positive response and 9 negative responses. Some statistics of this corpus are presented in Table 1.

Douban conversation corpus (Wu et al., 2017) which are constructed from Douban group [3](a popular social networking service in China) is also used in experiments. Response candidates on the test set are collected by Lucene retrieval model, other than negative sampling without human judgment on Ubuntu Dialogue Corpus. That is, the last turn of each Douban dialogue with additional keywords extracted from the context on the test set was used as query to retrieve 10 response candidates from the Lucene index set (Details are referred to section 4 in (Wu et al., 2017)). For the

---

[2]https://github.com/rkadlec/ubuntu-ranking-dataset-creator
[3]https://www.douban.com/group

|                             | Train   | Validation | Test    |
| --------------------------- | ------- | ---------- | ------- |
| #positive pairs             | 499,873 | 19,560     | 18,920  |
| #negative pairs             | 500,127 | 176,040    | 170,280 |
| #context                    | 957,119 | 19,560     | 18,920  |
| Median #tokens in contexts  | 64      | 66         | 68      |
| Median #tokens in responses | 13      | 14         | 14      |

Table 1: Statistics of the Ubuntu Dialogue Corpus (V2).

performance measurement on test set, we ignored samples with all negative responses or all positive responses. As a result, 6,670 context-response pairs were left on the test set. Some statistics of Douban conversation corpus are shown below:

|                               | Train | Validation | Test  |
| ----------------------------- | ----- | ---------- | ----- |
| #context-response pairs       | 1M    | 50k        | 10k   |
| #candidates per context       | 2     | 2          | 10    |
| #positive candidates per context | 1  | 1          | 1.18  |
| Avg. # turns per context      | 6.69  | 6.75       | 6.45  |
| Avg. #words per utterance     | 18.56 | 18.50      | 20.74 |

Table 2: Statistics of Douban Conversation Corpus (Wu et al., 2017).

## 4.2 IMPLEMENTATION DETAILS

Our model was implemented based on Tensorflow (Abadi et al., 2016). ADAM optimization algorithm (Kingma & Ba, 2014) was used for training. The initial learning rate was set to 0.001 and exponentially decayed during the training [4]. The batch size was 128. The number of hidden units of biLSTM for character-level embedding was set to 40. We used 200 hidden units for both context representation layers and matching aggregation layers. In the prediction layer, the number of hidden units with ReLu activation was set to 256. We did not use dropout and regularization.

Word embedding matrix was initialized with pre-trained 300-dimensional GloVe vectors [5] (Pennington et al., 2014). For character-level embedding, we used one hot encoding with 69 characters (68 ASCII characters plus one unknown character). Both word embedding and character embedding matrix were fixed during the training. After algorithm 1 was applied, the remaining out-of-vocabulary words were initialized as zero vectors. We used Stanford PTBTokenizer (Manning et al., 2014) on the Ubuntu corpus. The same hyper-parameter settings are applied to both Ubuntu Dialogue and Douban conversation corpus. For the ensemble model, we use the average prediction output of models with different runs. On both corpuses, the dimension of word2vec vectors generated on the training set is 100.

## 4.3 OVERALL RESULTS

Since the output scores are used for ranking candidates, we use Recall@k (recall at position k in 10 candidates, denotes as R@1, R@2 below), P@1 (precision at position 1), MAP(mean average precision) (Baeza-Yates et al., 1999), MRR (Mean Reciprocal Rank) (Voorhees et al., 1999) to measure the model performance. Table 3 and Table 4 show the performance comparison of our model and others on Ubuntu Dialogue Corpus V2 and Douban conversation corpus, respectively.

On Douban conversation corpus, FastText (Bojanowski et al., 2016) pre-trained Chinese embedding vectors [6] are used in ESIM + enhanced word vector whereas word2vec generated on training set is used in baseline model (ESIM). It can be seen from table 3 that character embedding enhances

---

[4] see tensorflow tf.train.exponential_decay
[5] glove.42B.300d.zip in https://nlp.stanford.edu/projects/glove/
[6] https://github.com/facebookresearch/fastText/blob/master/pretrained-vectors.md

| Model | 1 in 10 R@1 | 1 in 10 R@2 | 1 in 10 R@5 | MRR |
|---|---|---|---|---|
| TF-IDF (Lowe et al., 2017) | 0.488 | 0.587 | 0.763 | - |
| Dual Encoder w/RNN (Lowe et al., 2017) | 0.379 | 0.561 | 0.836 | - |
| Dual Encoder w/LSTM (Lowe et al., 2017) | 0.552 | 0.721 | 0.924 | - |
| RNN-CNN (Baudiš et al., 2016) | 0.672 | 0.809 | 0.956 | 0.788 |
| * MEMN2N (Dodge et al., 2015) | 0.637 | - | - | - |
| * CNN + LSTM(Ensemble) (Kadlec et al., 2015) | 0.683 | 0.818 | 0.957 | - |
| * Multi-view dual Encoder (Zhou et al., 2016) | 0.662 | 0.801 | 0.951 | - |
| * $\text{SMN}_{dynamic}$ (Wu et al., 2017) | 0.726 | 0.847 | 0.961 | - |
| ESIM | 0.696 | 0.820 | 0.954 | 0.802 |
| ESIM + char embedding | 0.717 | 0.839 | 0.964 | 0.818 |
| $\text{ESIM}^a$ (single) | 0.734 | 0.854 | 0.967 | 0.831 |
| $\text{ESIM}^a$ (ensemble) | 0.759 | 0.872 | 0.973 | 0.848 |

Table 3: Performance of the models on Ubuntu Dialogue Corpus V2. $\text{ESIM}^a$: ESIM + character embedding + enhanced word vector. Note: * means results on dataset V1 which are not directly comparable.

| Model | P@1 | MAP | MRR |
|---|---|---|---|
| TF-IDF (Wu et al., 2017) | 0.180 | 0.331 | 0.359 |
| RNN (Wu et al., 2017) | 0.208 | 0.390 | 0.422 |
| CNN (Wu et al., 2017) | 0.226 | 0.417 | 0.440 |
| LSTM (Wu et al., 2017) | 0.320 | 0.485 | 0.527 |
| BiLSTM (Wu et al., 2017) | 0.313 | 0.479 | 0.514 |
| Multi-View (Zhou et al., 2016; Wu et al., 2017) | 0.342 | 0.505 | 0.543 |
| DL2R (Yan et al., 2016a; Wu et al., 2017) | 0.330 | 0.488 | 0.527 |
| MV-LSTM (Wan et al., 2016; Wu et al., 2017) | 0.348 | 0.498 | 0.538 |
| Match-LSTM (Wang & Jiang, 2015; Wu et al., 2017) | 0.345 | 0.500 | 0.537 |
| Attentive-LSTM (Tan et al., 2015; Wu et al., 2017) | 0.331 | 0.495 | 0.523 |
| Multi-Channel (Wu et al., 2017) | 0.349 | 0.506 | 0.543 |
| $\text{SMN}_{dynamic}$ (Wu et al., 2017) | 0.397 | 0.529 | 0.569 |
| ESIM | 0.407 | 0.544 | 0.588 |
| ESIM + enhanced word vector (single) | 0.433 | 0.559 | 0.607 |

Table 4: Performance of the models on Douban Conversation Corpus.

the performance of original ESIM. Enhanced Word representation in algorithm 1 improves the performance further and has shown that the proposed method is effective. Most models (RNN, CNN, LSTM, BiLSTM, Dual-Encoder) which encode the whole context (or response) into compact vectors before matching do not perform well. $\text{SMN}_{dynamic}$ directly models sequential structure of multi utterances in context and achieves good performance whereas ESIM implicitly makes use of end-of-utterance(‗‗eou‗‗) and end-of-turn (‗‗eot‗‗) token tags as shown in subsection 4.6.

### 4.4 EVALUATION OF SEVERAL WORD EMBEDDING REPRESENTATIONS

In this section we evaluated word representation with the following cases on Ubuntu Dialogue corpus and compared them with that in algorithm 1.

WP1 Used the fixed pre-trained GloVe vectors [7].

WP2 Word embedding were initialized by GloVe vectors and then updated during the training.

WP3 Generated word2vec embeddings on the training set (Mikolov et al., 2013) and updated them during the training (dropout).

WP4 Used the pre-built ConceptNet NumberBatch (Speer et al., 2017) [8].

---

[7] glove.42B.300d.zip in https://nlp.stanford.edu/projects/glove/
[8] numberbatch-en-17.06.txt.gz in https://github.com/commonsense/conceptnet-numberbatch

WP5  Used the fixed pre-built FastText vectors [9] where word vectors for out-of-vocabulary words were computed based on built model.

WP6  Enhanced word representation in algorithm 1.

We used gensim [10] to generate word2vec embeddings of dim 100.

| Model | 1 in 10 R@1 | 1 in 10 R@2 | 1 in 10 R@5 | MRR |
|---|---|---|---|---|
| ESIM + char embedding (WP1) | 0.717 | 0.839 | 0.964 | 0.818 |
| ESIM + char embedding (WP2) | 0.698 | 0.824 | 0.956 | 0.805 |
| ESIM + char embedding (WP3) | 0.708 | 0.836 | 0.962 | 0.813 |
| ESIM + char embedding (WP4) | 0.706 | 0.831 | 0.962 | 0.811 |
| ESIM + char embedding (WP5) | 0.719 | 0.840 | 0.962 | 0.819 |
| ESIM + char embedding (WP6) | 0.734 | 0.854 | 0.967 | 0.831 |

Table 5: Performance comparisons of several word representations on Ubuntu Dialogue Corpus V2.

It can be observed that tuning word embedding vectors during the training obtained the worse performance. The ensemble of word embedding from ConceptNet NumberBatch did not perform well since it still suffers from out-of-vocabulary issues. In order to get insights into the performance improvement of WP5, we show word coverage on Ubuntu Dialogue Corpus.

| | Percent of #unique tokens | Percent of #tokens |
|---|---|---|
| Pre-trained GloVe vectors | 26.39 | 87.32 |
| Word2vec generated on training set | 8.37 | 98.8 |
| __eou__ and __eot__ | - | 10.9 |
| WP5 | 28.35 | 99.18 |

Table 6: Word coverage statistics of different word representations on Ubuntu Dialogue Corpus V2.

__eou__ and __eot__ are missing from pre-trained GloVe vectors. But this two tokens play an important role in the model performance shown in subsection 4.6. For word2vec generated on the training set, the unique token coverage is low. Due to the limited size of training corpus, the word2vec representation power could be degraded to some extent. WP5 combines advantages of both generality and domain adaptation.

## 4.5  EVALUATION OF ENHANCED REPRESENTATION ON A SIMPLE MODEL

In order to check whether the effectiveness of enhanced word representation in algorithm 1 depends on the specific model and datasets, we represent a doc (context, response or query) as the simple average of word vectors. Cosine similarity is used to rank the responses. The performances of the simple model on the test sets are shown in Figure 2.

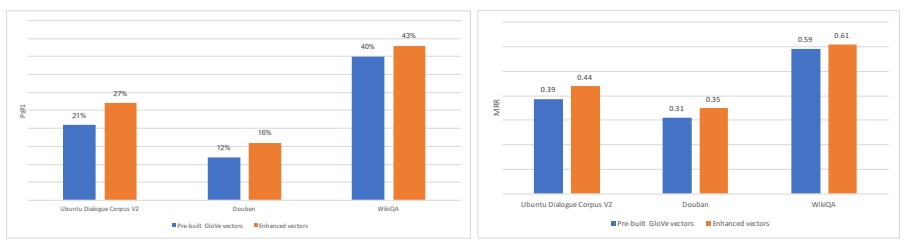

Figure 2: Performance comparisons of the simple average model.

[9]https://s3-us-west-1.amazonaws.com/fasttext-vectors/wiki.en.zip
[10]https://radimrehurek.com/gensim/

where WikiQA (Yang et al., 2015) is an open-domain question answering dataset from Microsoft research. The results on the enhanced vectors are better on the above three datasets. This indicates that enhanced vectors may fuse the domain-specific info into pre-built vectors for a better representation.

## 4.6 THE ROLES OF UTTERANCE AND TURN TAGS

There are two special token tags (__eou__ and __eot__) on ubuntu dialogue corpus. __eot__ tag is used to denote the end of a user's turn within the context and __eou__ tag is used to denote of a user utterance without a change of turn. Table 7 shows the performance with/without two special tags.

| Model | 1 in 10 R@1 | 1 in 10 R@2 | MRR |
|---|---|---|---|
| ESIM + char embedding (with eou and eot tags) | 0.717 | 0.839 | 0.818 |
| ESIM + char embedding (without eou and eot tags) | 0.683 | 0.812 | 0.793 |

Table 7: Performance comparison with/without __eou__ and __eot__ tags on Ubuntu Dialogue Corpus (V2).

It can be observed that the performance is significantly degraded without two special tags. In order to understand how the two tags helps the model identify the important information, we perform a case study. We randomly selected a context-response pair where model trained with tags succeeded and model trained without tags failed. Since max pooling is used in Equations 9 and 10, we apply max operator to each context token vector in Equation 7 as the signal strength. Then tokens are ranked in a descending order by it. The same operation is applied to response tokens.

It can be seen from Table 8 that __eou__ and __eot__ carry useful information. __eou__ and __eot__ captures utterance and turn boundary structure information, respectively. This may provide hints to design a better neural architecture to leverage this structure information.

| context | positive response |
|---|---|
| **Model with tags** | |
| i ca n't seem to get ssh to respect a changes(0.932) authorize_keys file __eou__(0.920) is there anything i should do besides service ssh restart ? __eou__(0.981) __eot__(0.957) restarting ssh should n't be necessary . . sounds like there 's(0.935) a different problem . are you sure the file is only readable by the owner ? and the . ssh directory is 700 ? __eou__ __eot__(0.967) | yeah , it was set up initially by ubuntu/ec2 , i(0.784) just changed(0.851) the file(0.837) , but it 's neither(0.802) locking out the old key(0.896) nor(0.746) accepting the new one __eou__ |
| **Model without tags** | |
| i ca n't seem to get ssh to respect a changes authorize_keys file is there anything i should do besides service ssh restart ? restarting(0.930) ssh should n't be necessary .(0.958) . sounds like there 's a different problem . are(0.941) you sure(0.935) the file(0.973) is only readable by the owner ? and the . ssh(0.949) directory is 700 ? | yeah , it was set up(0.787) initially by ubuntu/ec2 , i just changed the file(0.923) , but it 's neither locking(0.844) out the(0.816) old key nor(0.846) accepting(0.933) the new one |

Table 8: Tagged outputs from models trained with/without __eou__ and __eot__ tags. The top 6 tokens with the highest signal strength are highlighted in blue color. The value inside the parentheses is signal strength.

## 5 CONCLUSION AND FUTURE WORK

We propose an algorithm to combine pre-trained word embedding vectors with those generated on training set as new word representation to address out-of-vocabulary word issues. The experimental results have shown that the proposed method is effective to solve out-of-vocabulary issue and improves the performance of ESIM, achieving the state-of-the-art results on Ubuntu Dialogue Corpus and Douban conversation corpus. In addition, we investigate the performance impact of two special tags: end-of-utterance and end-of-turn. In the future, we may design a better neural architecture to leverage utterance structure in multi-turn conversations.

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
