# OpenReview forum: "Enhance Word Representation for Out-of-Vocabulary on Ubuntu Dialogue Corpus"
_ICLR.cc/2018/Conference — Reject_

### Official Review · AnonReviewer5 · 2017-11-24
**Good paper with important practical and engineering relevance. Little methodological novelty, though.**

**Rating:** 6
**Confidence:** 3

**Review:**

Summary:
This paper proposes an approach to improve the out-of-vocabulary embedding prediction for the task of modeling dialogue conversations. The proposed approach uses generic embeddings and combines them with the embeddings trained on the training dataset in a straightforward string-matching algorithm. In addition, the paper also makes a couple of improvements to Chen et. al's enhanced LSTM by adding character-level embeddings and replacing average pooling by LSTM last state summary vector. The results are shown on the standard Ubuntu dialogue dataset as well as a new Douban conversation dataset. The proposed approach gives sizable gains over the baselines.


Comments:

The paper is well written and puts itself nicely in context of previous work. Though, the proposed extension to handle out-of-vocabulary items is a simple and straightforward string matching algorithm, but nonetheless it gives noticeable increase in empirical performance on both the tasks. All in all, the methodological novelty of the paper is small but it has high practical relevance in terms of giving improved accuracy on an important task of dialogue conversation.

---

### Official Review · AnonReviewer2 · 2017-11-26
**A minor solution to resolving OOV word representations**

**Rating:** 3
**Confidence:** 5

**Review:**

The paper considers a setting (Ubuntu Dialogue Corpus and Douban Conversation Corpus) where most word types in the data are not covered by pretrained representations. The proposed solution is to combine (1) external pretrained word embeddings and (2) pretrained word embeddings on the training data by keeping them as two views: use the view if it's available, otherwise use a zero vector. This scheme is shown to perform well compared to other methods, specifically combinations of pretraining vs not pretraining embeddings on the training data, updating vs not updating embeddings during training, and others.

Quality: Low. The research is not very well modularized: the addressed problem has nothing specifically to do with ESIM and dialogue response classification, but it's all tangled up. The proposed solution is reasonable but rather minor. Given that the model will learn task-specific word representations on the training set anyway, it's not clear how important it is to follow this procedure, though minor improvement is reported (Table 5).

Clarity: The writing is clear. But the point of the paper is not immediately obvious because of its failure to modularize its contributions (see above).

Originality: Low to minor.

Significance: It's not convincing that an incremental improvement in the pretraining phase is so significant, for instance compared to developing a novel better architecture actually tailored to the dialogue task.

---

> ### Author Response · Authors · 2017-12-11
> **Reply to "a minor solution to resolving OOV word representations"**
>
> Thank for your valuable feedback.
>
> > the addressed problem has nothing specifically to do with ESIM and dialogue response classification, but it's all tangled up. The proposed solution is reasonable but rather minor.
>
> In order to check whether the effectiveness of the proposed enhanced representation depends on ESIM model and dataset, I uploaded a revision (12/11/2017) to use a very simple model (represent contexts/responses by a simple average of word vectors).  I evaluated it on Ubuntu, Douban and WikiQA datasets.  The results on the enhanced representation are still better on the above three datasets.  This may indicate that the enhanced vectors may fuse domain-specific info into pre-built vectors. Also this process is unsupervised.
>
> See section "4.5 EVALUATION OF ENHANCED REPRESENTATION ON A SIMPLE MODEL"

---

### Official Review · AnonReviewer4 · 2017-11-27
**Promising results but insufficient clarity and focus in write-up**

**Rating:** 5
**Confidence:** 4

**Review:**

The main contributions in this paper are:
1) New variants of a recent LSTM-based model ("ESIM") are applied to the task of response-selection in dialogue modeling -- ESIM was originally introduced and evaluated for natural language inference. In this new setting, the ESIM model (vanilla and extended) outperform previous models when trained and evaluated on two distinct conversational datasets.

2) A fairly trivial method is proposed to extend the coverage of pre-trained word embeddings to deal with the OOV problem that arises when applying them to these conversational datasets.
The method itself is to combine d1-dimensional word embeddings that were pretrained on a large unannotated corpus (vocabulary S) with distinct d2-dimensional word embeddings that are trained on the task-specific training data (vocabulary T). The enhanced (d1+d2)-dimensional representation for a word is constructed by concatenating its vectors from the two embeddings, setting either the d1- or d2-dimensional subvector to zeros when the word is absent from either S or T, respectively. This method is incorporated as an extension into ESIM and evaluated on the two conversation datasets.

The main results can be characterized as showing that this vocabulary extension method leads to performance gains on two datasets, on top of an ESIM-model extended with character-based word embeddings, which itself outperforms the vanilla ESIM model.

These empirical results are potentially meaningful and could justify reporting, but the paper's organization is very confusing, and too many details are too unclear, leading to low confidence in reproducibility.

There is basic novelty in applying the base model to a new task, and the analysis of the role of the special conversational boundary tokens is interesting and can help to inform future modeling choices. The embedding-enhancing method has low originality but is effective on this particular combination of model architecture, task and datasets. I am left wondering how well it might generalize to other models or tasks, since the problem it addresses shows up in many other places too...

Overall, the presentation switches back and forth between the Douban corpus and the Ubuntu corpus, and between word2vec and Glove embeddings, and this makes it very challenging to understand the details fully.

S3.1 - Word representation layer: This paragraph should probably mention that the character-composed embeddings are newly introduced here, and were not part of the original formulation of ESIM. That statement is currently hidden in the figure caption.

Algorithm 1:
- What set does P denote, and what is the set-theoretic relation between P and T?
- Under one possible interpretation, there may be items in P that are in neither T nor S, yet the algorithm does not define embeddings for those items even though its output is described as "a dictionary with word embeddings ... for P". This does not seem consistent? I think the sentence in S4.2 about initializing remaining OOV words as zeros is relevant and wonder if it should form part of the algorithm description?

S4.1 - What do the authors mean by the statement that response candidates for the Douban corpus were "collected by Lucene retrieval model"?

S4.2 - Paragraph two is very unclear. In particular, I don't understand the role of the Glove vectors here when Algorithm 1 is used, since the authors refer to word2vec vectors later in this paragraph and also in the Algorithm description.

S4.3 - It's insufficiently clear what the model definitions are for the Douban corpus. Is there still a character-based LSTM involved, or does FastText make it unnecessary?

S4.3 - "It can be seen from table 3 that the original ESIM did not perform well without character embedding." This is a curious way to describe the result, when, in fact, the ESIM model in table 3 already outperforms all the previous models listed.

S4.4 - gensim package -- for the benefit of readers unfamiliar with gensim, the text should ideally state explicitly that it is used to create the *word2vec* embeddings, instead of the ambiguous "word embeddings".

---

> ### Author Response · Authors · 2017-12-04
> **Reply to "Promising results but insufficient clarity and focus in write-up"**
>
> Thank for your feedback.  I have uploaded a new revision based on your suggestions.
>
> >  The embedding-enhancing method has low originality but is effective on this particular combination of model architecture, task and datasets. I am left wondering how well it might generalize to other models or tasks, since the problem it addresses shows up in many other places too...
>
> Good point. I will test embedding-enhanced method on other benchmark set/task to check whether it is still effective. I will report results here.
>
> > S3.1 - Word representation layer: This paragraph should probably mention that the character-composed embeddings are newly introduced here
> I updated it in revised version based on your advice.
>
> > What set does P denote, and what is the set-theoretic relation between P and T?
> P: all words in training/validation/testing sets (number of unique words could be large)
> T:  words with word2vec embedding on the training set.  T is a subset of P.  Word2vec also uses word document frequency to remove some low frequency words.
>
> In the revised version,  I change output  to " dimension d1 + d2 for (S\cap P) \cup T" and added notes "The remaining words which are in P and not in the above output dictionary are initialized with zero vectors".  Here we did not store word with zero vector in the above dictionary to save space in the output dictionary. This initialization is usually done during neural network initialization stage.
>
> > S4.1 - What do the authors mean by the statement that response candidates for the Douban corpus were "collected by Lucene retrieval model"?
> Based on your advice, I added the following sentences in the revised paper
> "That is, the last turn of each Douban dialogue with additional keywords extracted from the context on the test set was used as query to retrieve 10 response candidates from the Lucene index set (Details are referred to section 4 in (Wu et al., 2017))."
>
> Douban data was created by Wu et al., not by us (paper: https://arxiv.org/pdf/1612.01627.pdf,
> See section 4: Response Candidate retrieval and Section 5.2 Douban Conversation Corpus). On this dataset,  response negative candidates on the training/validation sets were random sampled whereas the retrieved method was used for testing set.
>
> > S4.2 - Paragraph two is very unclear. In particular, I don't understand the role of the Glove vectors here when Algorithm 1 is used, since the authors refer to word2vec vectors later in this paragraph and also in the Algorithm description.
>
> Here GloVe vectors are just pre-trainined word embedding ones from a general large dataset.
>
> For the clarification,  I added the following sentence  in Section 3.2
> "Here the pre-trainined word vectors can be from known methods such as GloVe (Pennington et al., 2014), word2vec (Mikolov et al., 2013) and FastText (Bojanowski et al., 2016).".
>
> On the training set we used word2vec in Algorithm 1 though other methods (GloVe and FastText) can be used too.
>
> > S4.3 - It's insufficiently clear what the model definitions are for the Douban corpus. Is there still a character-based LSTM involved,
> I used the same model layout and hyper-parameters for Douban and Ubuntu corpus.  In Section 4.2
> "The same hyper-parameter settings are applied to both Ubuntu Dialogue and Douban conversation corpus."
>
> Only the differences are pre-trained embedding vectors and word2vec generated on the training sets.   Wu et al's Douban dataset (Chinese) have been already tokenized so that it is easy for us to run word2vec based on gensim.
>
> > does FastText make it unnecessary?
> For western languages such as English, Germany, FastText generates ngram (character) internal embeddings and are used to address out-of-vocabulary issue.  For OOV (a word is out of FastText pre-trained embeddings), we can use average of word ngram to obtain its representation. For Ubuntu corpus, I can test it if you think that it is useful.
> For Douban, it is not easy for us to do it since dataset has been tokenized by Chinese tokenizer.
>
> > S4.3 - "It can be seen from table 3 that the original ESIM did not perform well without character embedding."
> Thanks.  I changed it to "
> It can be seen from table 3 that character embedding enhances the performance of original ESIM."
> "
>
> > S4.4 - gensim package -- for the benefit of readers unfamiliar with gensim, the text should ideally state explicitly that it is used to create the *word2vec* embeddings,
> I updated it in revised version based on your advice.

---

> > ### Author Response · Authors · 2017-12-07
> > **Reply to ""Promising results but insufficient clarity and focus in write-up"**
> >
> > I uploaded a new revision on Dec. 6.
> > On Table 5,  added performance comparison with FastText vectors.
> >
> > Used the fixed pre-built FastText vectors ( https://s3-us-west-1.amazonaws.com/fasttext-vectors/wiki.en.zip) where word vectors for out-of-vocabulary words were computed based on built model.
> > That is,
> > all_words_on_ubtuntu_dataset|./fasttext  print-word-vectors wiki.en.bin > ubuntu_fastText_word_vectors.txt
> > (see: https://github.com/facebookresearch/fastText)
> >
> > The performance of the proposed method is better.

---

> > > ### Author Response · Authors · 2017-12-11
> > > **Reply to "Promising results but insufficient clarity and focus in write-up"**
> > >
> > > I uploaded the revision on 12/11/2017 to address whether  the effectiveness of the proposed enhanced representation depends on ESIM model and datasets.
> > >
> > > I added a section "4.5 EVALUATION OF ENHANCED REPRESENTATION ON A SIMPLE MODEL". Here I used  a very simple model : represent contexts (or responses) by a simple average of word vectors. Cosine-similarity is used to rank candidate responses.  The results on the enhanced vectors are still better. I also tested it on WikiQA dataset.

---

### Public Comment · ~Alex_Wong1 · 2017-11-27
**Request for Code**

Dear Authors,

I am part of a team at McGill University participating in the ICLR 2018 Reproducibility Challenge (linked below). We have chosen to reproduce your study and are wondering if you would like to share some or all of the code you used.

http://www.cs.mcgill.ca/~jpineau/ICLR2018-ReproducibilityChallenge.html

Thank you!

---

> ### Author Response · Authors · 2017-11-28
> **Reply to "Request for Code"**
>
> We are extremely excited that you have selected our paper for reproducibility.  We are going through our employer's open source approval process which will take a much longer time than the Dec 15 deadline.  Few questions that may help us with alternatives.
>
> 1. Do we need to stay anonymous to continue further our correspondence?
> 2. Are you open for us to enter a legal contractual agreement to access our source code between our employer and your school for the "reproducibility" purpose?  This would be potentially a faster process to give you access to our source code.  I can explore this route to get more affirmative answers on the timing if you are open to enter a legal contractual agreement, like "no cost collaboration".
>
> At this time, we believe open source process would take beyond your Dec 15 deadline, but we hope to finish the open source approval for the conference date.
> If you have any further questions about our paper, please let us know as well.

---

> > ### Public Comment · ~Alex_Wong1 · 2017-11-29
> > **Reply to "Request for Code"**
> >
> > Thank you for your willingness to help!
> >
> > Unfortunately, we are not in a position to enter a legal contractual agreement on behalf of the University, but if there is still a way to share any source code anonymously that would be helpful. We would only be using your source material for this reproduction challenge. If not, then you probably do not need to stay anonymous for further correspondence. You can find my email address on my OpenReview profile.
> >
> > In terms of implementation details, we do have some questions:
> > 1. What random seed did you use to generate the Ubuntu corpus?
> > 2. How did you implement the character-composed embedding? More specifically, could you give more detail on you are describing in this line from section 3.1: "The character-composed embedding is generated by concatenating the final state vector of the forward and backward direction of bi-directional LSTM (BiLSTM)"
> > 3. Could you clarify on the concatenation of word and character embeddings?
> > 4. Regarding your ESIM, what settings did you use for the following hyper-parameters: patience, gradient clipping threshold, max epochs?
> >
> > Again, if you'd rather communicate through email or another channel, feel free.
> >
> > Thanks!

---

> > > ### Author Response · Authors · 2017-11-29
> > > **Reply to "Request for Code"**
> > >
> > > Hi, Alex,
> > >       I could not see your email in open review profile ("a****4@cs").  Open source the code in the paper  is in progress.  I don't know whether I can share the code for this reproduction challenge now and need to check the legal department in my company.
> > >
> > > > 1. What random seed did you use to generate the Ubuntu corpus?
> > > just used the default one (default = 1234) (see: https://github.com/rkadlec/ubuntu-ranking-dataset-creator) so that results are comparable with others.
> > >
> > > > 2. How did you implement the character-composed embedding?
> > > > 3. could you clarify on the concatenation of word and character embeddings?
> > >
> > > I used the tensorflow (tf.nn.bidirectional_dynamic_rnn)  to  conduct all experiments in the paper.
> > > For example, you can define function below:
> > > def lstm_layer(inputs, input_seq_len, rnn_size, dropout_keep_prob, scope, scope_reuse=False):
> > >     with tf.variable_scope(scope, reuse=scope_reuse) as vs:
> > >         fw_cell = tf.contrib.rnn.LSTMCell(rnn_size, forget_bias=1.0, state_is_tuple=True, reuse=scope_reuse)
> > >         fw_cell  = tf.contrib.rnn.DropoutWrapper(fw_cell, output_keep_prob=dropout_keep_prob)
> > >         bw_cell = tf.contrib.rnn.LSTMCell(rnn_size, forget_bias=1.0, state_is_tuple=True, reuse=scope_reuse)
> > >         bw_cell  = tf.contrib.rnn.DropoutWrapper(bw_cell, output_keep_prob=dropout_keep_prob)
> > >         rnn_outputs, rnn_states = tf.nn.bidirectional_dynamic_rnn(cell_fw=fw_cell, cell_bw=bw_cell,
> > >                                                                 inputs=inputs,
> > >                                                                 sequence_length=input_seq_len,
> > >                                                                 dtype=tf.float32)
> > >         return rnn_outputs, rnn_states
> > >
> > > Then
> > > #context_char_embedded: [batch_size * max_sequence_length, max_word_length, embed_char_dim]
> > > #context_char_length: [batch_size * max_sequence_length] (define number of character per word)
> > > #charRNN_size:  40
> > > #max_word_length: 18
> > > #max_sequence_length: 180
> > > #dropoutput_keep_prob: 1.0
> > > #embed_char_dim: 69
> > > #batch_size: 128
> > > char_rnn_output_context,  char_rnn_state_context = lstm_layer(context_char_embedded, context_char_length, charRNN_size,  dropout_keep_prob, charRNN_scope_name, scope_reuse=False)
> > >
> > > #response_char_embedded: [batch_size * max_sequence_length, max_word_length, embed_char_dim]
> > > #response_char_length: [batch_size * max_sequence_length]
> > >
> > > char_rnn_output_response, char_rnn_state_response = lstm_layer(response_char_embedded,
> > > response_char_length, charRNN_size,  dropout_keep_prob, charRNN_scope_name, scope_reuse=True)
> > >
> > > #context char representation
> > > char_embed_dim = charRNN_size * 2
> > > #context_char_state: [batch_size * max_sequence_length,  char_embed_dim]
> > > context_char_state = tf.concat(axis=1, values=[char_rnn_state_context[0].h, char_rnn_state_context[1].h])
> > > #reshape
> > > context_char_state = tf.reshape(context_char_state, [-1, max_sequence_length, char_embed_dim])
> > >
> > > The similar operations are applied to char_rnn_state_response.
> > >
> > > For word embedding, I assume that you can get "context_word_output and response_word_output"
> > > Both tensors will have shape [batch_size, max_sequence_length, word_embedding_dim]
> > > Then you can use tf.concat to get the combined representation.
> > >
> > > > Regarding your ESIM, what settings did you use for the following hyper-parameters: patience, gradient clipping threshold, max epochs?
> > > I am not familiar with patience.  No gradient clipping was used.  In my experiments, training usually achieved the best performance (MRR) on the validation set at around 22000 - 25000 batch steps.
> > >
> > > Note:  tensorflow version (tensorflow-gpu (1.1.0)).
> > >
> > > If you share your email, we can communicate through email or another channel.

---

> > > > ### Public Comment · ~Alex_Wong1 · 2017-11-30
> > > > **Reply to "Request for Code"**
> > > >
> > > > Hi,
> > > >
> > > > Thanks for all of this, will definitely take time to go through your notes. For more communication, my email is alexander.wong4@mail.mcgill.ca
> > > >
> > > > Cheers

---

> > > > ### Public Comment · ~Alex_Wong1 · 2017-12-07
> > > > **Further Clarifications**
> > > >
> > > > Hi,
> > > >
> > > > We have a few more questions.
> > > >
> > > > 1. We used starnford CoreNLP's library (https://stanfordnlp.github.io/CoreNLP/tokenize.html) to use the equivalent of the PTBTokenizer, as stated in the paper. However, this produced 811,059 tokens (instead of 823,057 tokens). Have you specified any special parameters when applying tokenization?
> > > > 2. Regarding word2vec, did you use any non-default hyperparameters? And for training, did you train contexts and responses as distinct inputs or concatenate the context-response pairs to train?
> > > >
> > > > We haven't received an email from you yet, but if you'd rather communicate through email, you can reach us at alexander.wong4@mail.mcgill.ca
> > > >
> > > > Thanks!

---

> > > > > ### Author Response · Authors · 2017-12-08
> > > > > **Reply to "Further clarification"**
> > > > >
> > > > > > 1. We used stanford CoreNLP's library
> > > > > We wrote a java program based on CoreNLP library to perform PTBTokenizer, other than command-line interface (CLI). For CLI, it is not easy to create input-output correspondence.
> > > > > See java API example (https://stanfordnlp.github.io/CoreNLP/api.html)
> > > > > Properties props = new Properties();
> > > > > props.put("annotators", "tokenize, ssplit, lemma"}
> > > > >
> > > > > > Regarding word2vec, did you use any non-default hyperparameters?
> > > > > use the default. Iter=20
> > > > > > did you train contexts and responses as distinct inputs or concatenate the context-response pairs to train?
> > > > > distinct inputs. Each context/response takes one line in the input data file.

---

> ### Author Response · Authors · 2017-11-28
> **Reply to "Request  for code"**
>
> Hi, Alex,
>      Thank for your interest in our paper. Open source approval process in our company may take time.  At the same time,  if further clarification about technical implementation details (e.g hyper-parameter setting) is needed, feel free to ask here. We like to help you reproduce the results in the paper.

---

### Public Comment · ~Hugo_Scurti1 · 2017-12-16
**Reproducibility Summary**

INTRODUCTION

As participants in the 2018 ICLR Reproducibility Challenge, we aimed to reproduce the findings of this paper. The paper presents a methodology to improve performance on modelling dialogue systems that contain out-of-vocabulary words. Ultimately, we implemented Chen et al.’s Enhanced LSTM Method (ESIM) to model the Ubuntu Dialogue Corpus V2.0 as presented in the paper.

Overall, we found the paper to be clear and concise. However, we found it difficult to implement the authors’ enhancements to the ESIM model. In particular, the training of character-composed embeddings is briefly described only as the concatenation of final state vectors at the BiLSTM. Further communication with the authors did not clarify enough for our purposes how exactly these embeddings are concatenated with word embeddings within the model. For this reason, as stated above, we decided only to replicate the ESIM.

REPRODUCTION

Downloading and preprocessing the Ubuntu Dialogue Corpus and pre-trained GloVe was a simple matter of following the procedure specified in the paper. Using publicly available datasets definitely facilitated reproducibility. In generating the dataset, all default parameters were used, with a random seed of ‘1234’ that the authors provided upon enquiry.

The generated data was modified into formats appropriate for Word2Vec and ESIM inputs. The dataset was tokenized using the publicly available Stanford CoreNLP library PTB Tokenizer, and then lemmatized. We then used a stored set of distinct tokens to filter the pre-trained GloVe vector, removing all words that do not appear in the training corpus. We thought this step would be beneficial since the unzipped glove dataset (glove.42B.300d.txt) is 4.67 GB large, which would take a considerable amount of memory simply to load it. The filtered GloVe dataset takes about 440 MB and contains roughly 9% of the original GloVe dataset. Through this process, we confirmed the authors’ observation that only 22% of the 823,057 Ubuntu tokens occur in the pre-built GloVe word vectors, and that our reproduction produced the same dataset.

To reproduce the baseline ESIM model, we were not able to access the source code of the paper’s authors due to issues regarding their employer’s open source policy. Instead, we implemented the ESIM using source code of an implementation by Williams et al., that was found on GitHub. We followed all hyperparameters specifications possible, and when particular hyperparameters were not provided, we consulted the authors who provided further detail. Specifically, these hyperparameters were ‘patience’, and ‘gradient clipping threshold’, and ‘max epochs’. Of these, the authors stated that ‘patience’ and ‘gradient clipping’ were not used, and that “training usually achieved the best performance (MRR) on the validation set at around 22000 - 25000 batch steps.” In general, the authors replied quickly and comprehensively to our enquiries within the comments section of OpenReview, which contributed positively to the reproducibility of the paper.

When training the model, performance metrics were printed every 50 steps. Accuracy and cost over the validation set and over a subset of the training set were employed to evaluate the training of the model. We did not evaluate over the whole training set since this is significantly larger and would greatly slower the training time. We implemented our own algorithms to evaluate R@k and MRR.

The paper does not detail the computing infrastructure that was used. For our implementation we used an Google Cloud Engine instance (full technical specifications in the linked report).

We were only able to train the model to 9750 steps given our implementation architecture. This took 65 hours to train. This is considerably less than the 22000-25000 steps described by the authors as providing the best results. It is very likely that the authors trained their model using multiple GPU units, which would be unfeasible for us given the cost of GPUs on a virtual machine.

RESULTS AND CONCLUSION

Given our limited computation power, we were able to train our reproduction model to the same level of performance as described by the authors. We observed a MRR of 0.733, lower than the MRR of 0.802 seen in the paper in question. However, on account of the methodology followed in our investigation, we can confirm that the model is re-computable. It seems highly likely that the paper’s results are valid, and would have been observed by us had our model been trained with more iterations.

The full report by H. Scurti, I. Sultan, and A. Wong is contained in the folder ‘report’ in the repository linked  below:

https://bitbucket.org/hugoscurti/comp551_f17_a4/src/

---

> ### Author Response · Authors · 2017-12-16
> **Reply to "Reproducibility Summary"**
>
> Thank Hugo et al very much for reproducing the results.
>
> > The paper does not detail the computing infrastructure that was used.
> Local machine :  Intel(R) Core(TM) i7-6700 CPU @ 3.40GHz  * 2
>                                RAM: 32G    ( 8  * 4G (DDR4, 2133 MHz)
>                                One GPU : nvidia P5000 (16 G GPU RAM)
>
> You used Telsa K80 (with 24G GPU RAM). I have not compared the performance between P5000 and Tesla K80.
>
> > Accuracy and cost over the validation set and over a subset of the training set were employed to evaluate the training of the model.
>
> In our experiments,  we evaluated the accuracy,  MRR, P@1 on the validation set every 1000 steps and saved the model with the highest MRR.  In your code, you saved the model with the best accuracy on the validation set every 50 steps.
> My suggestion:
>     1) use MRR
>     2) perform the evaluation on the validation set every K steps (K could be larger to reduce the computational cost since evaluation on the validation set is slow). This will help you speed up the training.
>
> > In training the character embeddings using Word2Vec,
> >we used all the default hyperparameters, and trained each
> > context/response as distinct inputs such that each context/response
> >pair takes one line in the input data file.
>
> I assume that there is a typo here.  'character embedding' may be 'word embedding'.
> In our algorithm 1, we used Word2vec to generate word embedding on the training set and concatenated them with pre-built GloVe vectors. Character Embedding is used in our ESIM. Since you only evaluated the baseline ESIM model, character embedding would not be used.
>
> >  the training of character-composed embeddings is briefly described only as the concatenation of final state vectors at the BiLSTM.
> The implementation of character embedding was showed in my first comment. It is relatively easy to integrate them into your code (see: tf_esim.py  Line 43 and Line 44). Character-embedding may consume more memory.

---

> ### Author Response · Authors · 2017-12-16
> **Reply to "Reproducibility Summary"**
>
> For reference and result reproducibility (ESIM^a in Table 3 in the paper),  I pasted the logs of performance evaluation on the validation every 1000 steps during the training.  It took about 13 hours 41 minutes to reach 23000 training steps.
>
> step: 1000
> MAP (mean average precision: 0.735673771383	MRR (mean reciprocal rank): 0.735673771383	Top-1 precision: 0.607566462168	Num_query: 19560
>
> Step: 2000
> MAP (mean average precision: 0.762894553186	MRR (mean reciprocal rank): 0.762894553186	Top-1 precision: 0.643149284254	Num_query: 19560
>
> Step: 3000
> MAP (mean average precision: 0.781005473594	MRR (mean reciprocal rank): 0.781005473594	Top-1 precision: 0.666462167689	Num_query: 19560
>
> Step: 4000
> MAP (mean average precision: 0.791324840945	MRR (mean reciprocal rank): 0.791324840945	Top-1 precision: 0.679396728016	Num_query: 19560
>
> Step: 5000
> MAP (mean average precision: 0.793004146785	MRR (mean reciprocal rank): 0.793004146785	Top-1 precision: 0.680112474438	Num_query: 19560
>
> Step: 6000
> MAP (mean average precision: 0.806250669491	MRR (mean reciprocal rank): 0.806250669491	Top-1 precision: 0.698108384458	Num_query: 19560
>
> ....
> Step: 9000
> MAP (mean average precision: 0.819590433992	MRR (mean reciprocal rank): 0.819590433992	Top-1 precision: 0.717791411043	Num_query: 19560
>
> Step: 10000
> MAP (mean average precision: 0.818069269971	MRR (mean reciprocal rank): 0.818069269971	Top-1 precision: 0.714008179959	Num_query: 19560
>
> Step: 11000
> MAP (mean average precision: 0.818855596942	MRR (mean reciprocal rank): 0.818855596942	Top-1 precision: 0.714979550102	Num_query: 19560
>
> Step: 12000
> MAP (mean average precision: 0.821677885708	MRR (mean reciprocal rank): 0.821677885708	Top-1 precision: 0.719325153374	Num_query: 19560
>
> Step: 13000
> MAP (mean average precision: 0.8232087472	MRR (mean reciprocal rank): 0.8232087472	Top-1 precision: 0.721523517382	Num_query: 19560
>
> Step: 14000
> MAP (mean average precision: 0.825161326971	MRR (mean reciprocal rank): 0.825161326971	Top-1 precision: 0.724948875256	Num_query: 19560
>
> Step: 15000
> MAP (mean average precision: 0.825991109975	MRR (mean reciprocal rank): 0.825991109975	Top-1 precision: 0.725051124744	Num_query: 19560
>
> Step: 16000
> MAP (mean average precision: 0.824983891648	MRR (mean reciprocal rank): 0.824983891648	Top-1 precision: 0.722750511247	Num_query: 19560
>
> Step: 17000
> MAP (mean average precision: 0.827094653812	MRR (mean reciprocal rank): 0.827094653812	Top-1 precision: 0.727198364008	Num_query: 19560
>
> Step: 18000
> MAP (mean average precision: 0.829552151297	MRR (mean reciprocal rank): 0.829552151297	Top-1 precision: 0.730981595092	Num_query: 19560
>
> Step: 19000
> MAP (mean average precision: 0.830157512903	MRR (mean reciprocal rank): 0.830157512903	Top-1 precision: 0.73200408998	Num_query: 19560
>
> Step: 20000
> MAP (mean average precision: 0.82902826468	MRR (mean reciprocal rank): 0.82902826468	Top-1 precision: 0.729703476483	Num_query: 19560
>
> Step: 21000
> MAP (mean average precision: 0.832002669848	MRR (mean reciprocal rank): 0.832002669848	Top-1 precision: 0.734918200409	Num_query: 19560
>
> Step: 22000
> MAP (mean average precision: 0.830050982731	MRR (mean reciprocal rank): 0.830050982731	Top-1 precision: 0.731339468303	Num_query: 19560
>
> Step: 23000
> MAP (mean average precision: 0.832678571429	MRR (mean reciprocal rank): 0.832678571429	Top-1 precision: 0.735736196319	Num_query: 19560
>
> Step: 24000
> MAP (mean average precision: 0.828641116467	MRR (mean reciprocal rank): 0.828641116467	Top-1 precision: 0.728936605317	Num_query: 19560
>
> Step: 25000
> MAP (mean average precision: 0.826601259454	MRR (mean reciprocal rank): 0.826601259454	Top-1 precision: 0.725766871166	Num_query: 19560

---

### Decision · Program_Chairs · 2018-01-29
**ICLR 2018 Conference Acceptance Decision**

**Decision:**

Reject

**Comment:**

This paper's idea is to augment pre-trained word embeddings on a large corpus with embeddings learned on the data of interest. This is shown to yield better results than the pre-trained word embeddings alone. This contribution is too limited to justify publication at iclr.